# Efficient Photoelectrochemical Water Splitting Reaction using Electrodeposited Co$_3$Se$_4$ Catalyst

**Yelyn Sim †, Jude John †, Subramani Surendran †, Byeolee Moon and Uk Sim ***

Department of Materials Science & Engineering, Chonnam National University, Gwangju 61186, Korea;
simyelyn0804@gmail.com (Y.S.); judejohnkarokaran@gmail.com (J.J.); surenj503@gmail.com (S.S.)
byeolee89@gmail.com (B.M.)
* Correspondence: usim@jnu.ac.kr; Tel.: +82-62-530-1718
† These authors contributed equally to this work.

**Abstract:** Photoelectrochemical water splitting is a promising field for sustainable energy production using hydrogen. Development of efficient catalysts is essential for resourceful hydrogen production. The most efficient catalysts reported to date have been extremely precious rare-earth metals. One of the biggest hurdles in this research area is the difficulty of developing highly efficient catalysts comparable to the noble metal catalysts. Here, we report that non-noble metal dichalcogenide (Co$_3$Se$_4$) catalysts made using a facile one-pot electrodeposition method, showed highly efficient photoelectrochemical activity on a Si photocathode. To enhance light collection and enlarge its surface area even further, we implemented surface nanostructuring on the Si surface. The nanostructured Si photoelectrode has an effective area greater than that of planar silicon and a wider absorption spectrum. Consequently, this approach exhibits reduced overvoltage as well as increased photo-catalytic activity. Such results show the importance of controlling the optimized interface between the surface structure of the photoelectrode and the electrodeposited co-catalyst on it to improve catalytic activity. This should enable other electrochemical reactions in a variety of energy conversion systems.

**Keywords:** photoelectrochemical cell; hydrogen evolution reaction (HER); metal free catalyst; cobalt selenide catalyst

## 1. Introduction

Increasing energy demand and the rapid depletion of fossil fuels have made it inevitable that renewable sources of alternative energy be found [1]. Despite the difficulties of finding alternative sources of energy, hydrogen has emerged as an excellent source of clean energy that could serve as a key technology for providing alternative energy [2]. The state-of-the-art high-efficiency hydrogen-producing catalysts are based on noble metals like Pt [3–5]. However, the cost and scarcity of noble metals are preventing hydrogen energy from being used in a variety of applications. Moreover, depletion of fossil fuels at a steady pace is contributing to the emerging energy crisis [6]. In this situation, hydrogen has become attractive as an excellent option for clean energy that could also serve as a key technology in providing sustainable alternative energy [7]. The discovery of photoelectrochemical water splitting using TiO$_2$ by Honda and Fujishima in 1972 has completely revolutionized the area of photoelectrochemical water splitting using semiconductor photoelectrodes [8]. In this regard, photoelectrochemical (PEC) water splitting to produce hydrogen is turning out to be a promising strategy through which energy produced from solar irradiation could be transformed and stored in the form of hydrogen [2,9].

Photoelectrodes provide the essential function of converting solar energy-to-fuel. To achieve this conversion, the photoelectrode should possess a reasonable band gap for absorption of the incoming incident light. This is necessary to drive the reaction for the reduction and oxidation of water. The desired photoelectrode also has to provide charge transfer kinetics fast enough to minimize the recombination of oppositely charged ions at the electrode/electrolyte interface [9]. Metal oxides have larger bandgaps (higher than 2 eV) which restrains their light absorption capability and thereby makes them inefficient for solar energy conversion [10–12]. Among the various semiconductor materials that have been researched, p-type silicon (Si) stands out as a well-known photocathode owing to its excellent photoelectrochemical water splitting properties [13–16]. Silicon is also an Earth-abundant semiconductor material with a narrow bandgap (1.12 eV). These qualities make it an appropriate material for absorbing a larger portion of the solar spectrum and thus an ideal candidate for the photovoltaic industry [17]. Even with these advantages, Si is quite unstable in aqueous conditions and in the absence of a catalyst, the reaction at the p-Si/electrolyte interface results in sluggish kinetics at zero overpotential (i.e., 0 V vs. RHE) [18].

One strategy to overcome this was to incorporate a corrosion-resistant protective layer that would minimize inhibition of the photogenerated carriers and thereby improve the charge transfer. Numerous studies have been carried out in the area of adding protective layers to Si photoelectrodes to improve the stability of p-Si photoelectrodes in aqueous conditions [19–21]. Platinum (Pt) [22] is well known for its use as an electrocatalyst in PEC hydrogen production. It is however, also a precious metal, the scarcity of which makes its large-scale deployment for PEC water splitting rather unlikely [23]. In response to these conditions, tremendous effort has been expended to find and develop Earth-abundant non-precious catalysts for hydrogen production.

Recently, first-row transition-metal dichalcogenides have emerged as significant prospects for use in water-splitting reactions and been proposed as replacements for precious metals (i.e., Pt). This proposal is especially compelling due to their natural-abundance and high catalytic activity. Among these options, $CoX_2$ (X = S, Se) has lower overpotential with smaller Tafel slope and large exchange current and variants have shown promising development in HER (hydrogen evolution reaction) reactions [24–26]. Yu et al. established that higher catalytic rendition could be achieved by turning bulk $CoSe_2$ into an ultrathin nanosheet and thereby significantly enhancing the surface-to-volume ratio due to the vacancy rich surfaces [27,28]. Similarly Zhang et al. found that by a phase engineering process, they were able to produce more edge sites on $CoSe_2$ catalyst. This included polymorphic phases of $CoSe_2$ that were obtained through a temperature controlled process for calcination of amorphous $CoSe_x$. Their work demonstrated that the grain boundaries existing between the two phases would unravel new HER active sites, thereby enhancing the overall HER activity in $CoSe_2$ [29].

However, regarding actual development of HER photocathodes, relatively few catalysts have been successfully integrated into these photocathodes. Moreover, many of these could not achieve their full potential due to the lack of effective interfacing of these active nanosized HER catalysts with the silicon photoabsorbers [30–33]. In the light of these developments, we report a simple and facile method for electrodeposition of $Co_3Se_4$ as a co-catalyst passivation-layer onto a p-Si photocathode to enhance the photoelectrochemical water-splitting reaction. To provide better substrate selectivity, the catalytic activity of the $Co_3Se_4$ co-catalyst was tested using Fluorine-doped Tin Oxide (FTO) and a glassy carbon electrode as substrates. The $Co_3Se_4$ decorated onto the FTO substrate exhibited excellent HER ($\eta_{(10)}$ = 215 mV) catalytic activity. The resulting low Tafel value 42.5 mV/dec occurred under pH 0 condition. Hence, new $Co_3Se_4$ electrocatalysts with optimized working electrode selectivity and controlled deposition rate are proposed as an efficient catalyst for water-splitting applications.

## 2. Materials and Methods

### 2.1. Materials

Sodium selenite ($Na_2SeO_3$), cobalt acetate ($Co(CH_3COO)_2$), lithium chloride (LiCl), perchloric acid ($HClO_4$), silver nitrate ($AgNO_3$), hydrochloric acid (HCl), nitric acid ($HNO_3$) and hydrofluoric acid (HF) were purchased from Sigma Aldrich (Seoul, Republic of Korea, 2018). All reagents used in this work were of analytical grade purity and were used without further purification. As (photo)electrodes, Si substrates (B-doped, p-type, 500 μm thickness, 10–15 Ω·cm resistivity, (100) oriented) were purchased from Namkang Inc. (Seongnam, Rep. of Korea, 2018) and FTO glass slides ($1 \times 2$ cm$^2$, 7 Ω/sq resistivity) were purchased from Wooyang GMS (Namyangju, Republic of Korea, 2018). Si wafers were cut to dimensions of $8 \times 8$ mm$^2$ and then each of these Si wafers was cleaned prior to deposition with acetone, 2-propanol and deionized water, respectively, using a bath sonicator (10 min each). In the chemical etching of the nanostructured Si (NSi) surface, the Si substrates were dipped into a solution containing 20 mL aqueous etchant solution of $AgNO_3$ (0.679 g, 0.02 M) and HF (5 M) under ambient conditions (25 °C). Finally, Ag residue was completely removed from the nanostructured silicon surface by treating with 70% nitric acid solution for two hours.

### 2.2. Electrodeposition of Co$_3$Se$_4$

Prior to electrodeposition, the substrates were cleaned using a standard protocol, which involves treating the substrates with acetone, 2-propanol and deionized water, respectively, using a bath sonicator (10 min each). For electrodeposition, substrates of planar silicon, nanostructured Si (NSi) and FTO were used to get the best possible results from the three electrodes. An aqueous solution containing 20mM $Na_2SeO_3$ (0.101 g), 20 mM $Co(CH_3COO)_2$ (0.146 g) and 100 mM LiCl (0.0121 g) were placed in a 30 mL container. The solution was kept near pH 2 by adding dilute HCl. The deposition was conducted at room temperature in a single compartment glass cell using a three-electrode configuration. Electrodeposition was performed with a CHI 7008E (CH Instruments, Inc., Austin, TX, USA, 2018) potentiostat. Graphite rod was used as a counter electrode and Ag/AgCl (3M NaCl) was used as the reference electrode for the electrodeposition. The chronoamperometric deposition was conducted at $-0.8$ V for varying deposition times (5, 10, or 15 s) and we determined that the optimal time was 15 s. A thin film of $Co_3Se_4$ co-catalyst was formed on the surface of the working electrodes.

### 2.3. Photoelectrochemical Measurements

Photoelectrochemical tests were conducted using a three-electrode cell. Working electrodes comprising a planar silicon photoelectrode, nanowire structured silicon photoelectrode and FTO glass were used. Graphite rods were used as counter electrodes and Ag/AgCl (3M NaCl) as reference electrodes. Each reference electrode was calibrated to $-0.201$ V (vs. RHE) in a proton-rich aqueous solution of 1 M perchloric acid and purged with a high-purity saturated $H_2$ at 25 °C. For solar irradiation, a 300 W Xenon lamp was used as the light source at a light intensity of 100 mW/cm$^2$ (Air Mass 1.5 Global condition glass filter).

## 3. Results and Discussion

Cobalt selenide ($Co_3Se_4$) was deposited electrochemically onto conductive electrodes by the electrodeposition method. The X-ray diffraction (XRD) pattern shown in Figure 1 reveals highly intense and crystalline peaks of prepared cobalt selenide sample. The peaks confirm the single-phase formation of pure monoclinic crystal system of $Co_3Se_4$, which exactly corresponds with the standard JCPDS card No. 98-009-9990. The prepared $Co_3Se_4$ was in well agreement with the lattice parameters of space group of P21/c (space group number 14). Field-emission scanning electron microscopy (FESEM) measurements were employed to understand the surface morphology of the $Co_3Se_4$ film on a glassy carbon electrode (GC) and a fluorine-doped tin oxide (FTO) as shown in Figure 1b,c. The morphology of $Co_3Se_4$ film has almost same 3D porous structures despite of the different electrode

substrates. The electrodeposition time was increased from 5 s to 25 s and confirmed that almost same 3D porous structure of $Co_3Se_4$ film among samples of 15 s, 20 s and 25 s of electrodeposition time. Representative samples on GC electrode were selected with samples of the electrodeposition time of 20 s (Figure 1b) and 25 s (the inset image in Figure 1b), respectively. Meanwhile, as shown in Figure 1b,c, the morphologies of $Co_3Se_4$ films on GC electrode are almost same with the $Co_3Se_4$ film on FTO with the electrodeposition time of 15 s in Figure 1d, showing the top-view and cross-sectional FESEM view images. As shown in Figure 1d, nanostructured $Co_3Se_4$ film was uniformly deposited with 3D porous structure. To determine the elemental composition of the film, energy dispersive spectrometry (EDS) was performed (Figure 1d). To clarify this further, transmission electron microscopy (TEM) measurements of the deposited film were conducted as shown in Figure 1f. Information about the lattice fringe with an inter-planar spacing of 2.67 nm, revealed that the electro-deposited $Co_3Se_4$ film strongly corresponds to the (111) plane of the obtained XRD pattern.

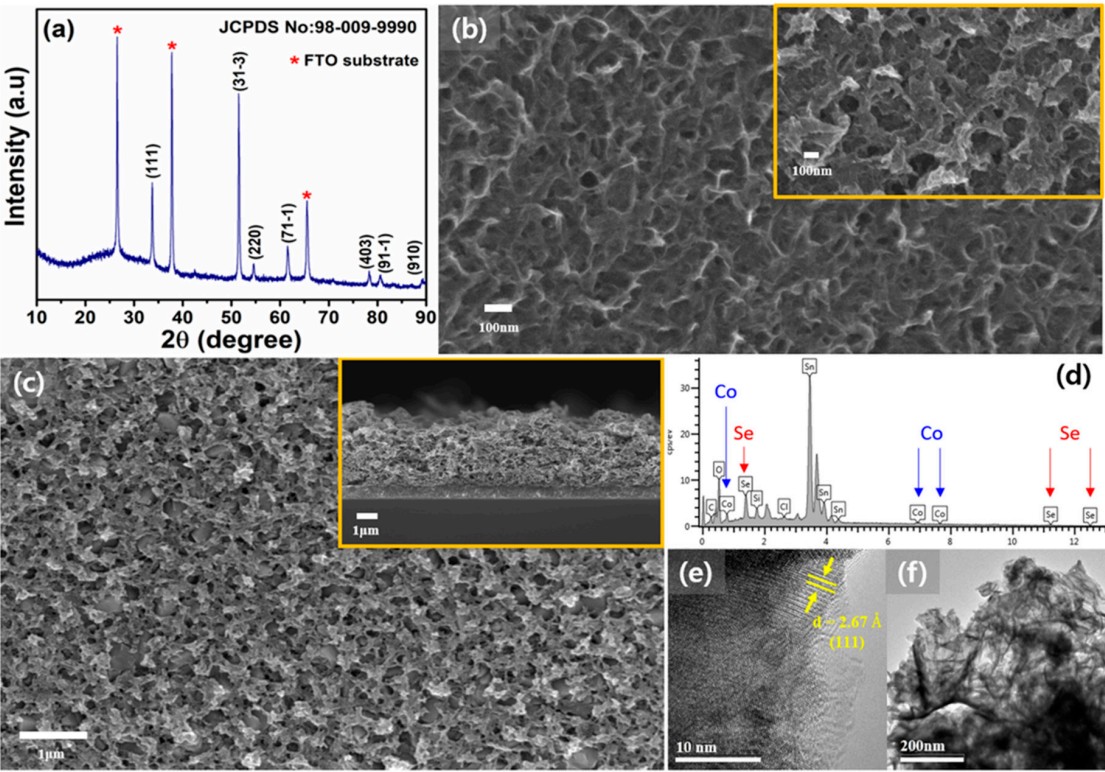

**Figure 1.** Surface characterization of the $Co_3Se_4$ film: (**a**) XRD (x-ray diffraction) pattern of electrodeposited $Co_3Se_4$ on FTO (fluorine doped tin oxide). (**b**) Plan-view FESEM (field-emission scanning electron microscope) images of electrodeposited $Co_3Se_4$ on GC electrode. (**c**) Plan-view and cross-sectional (inset) FESEM images of electrodeposited $Co_3Se_4$ on FTO electrode. Different but conducting electrode substrates result in the same 3D porous structure of $Co_3Se_4$ film through the electrodeposition process. (**d**) Energy dispersive spectrometry spectrum showing the presence of cobalt and selenide elements and (**e**) High-resolution transmission electron microscopy (HR-TEM) image to identify $Co_3Se_4$. (**f**) TEM (transmission electron microscopy) image showing uniformly deposited $Co_3Se_4$.

Photoelectrochemical measurements indicated that the electrodeposited $Co_3Se_4$ film on the photoelectrode acted as an effective layer for the hydrogen evolution reaction (HER). Figure 2a shows the resulting photoresponse of bare planar Si and $Co_3Se_4$ on planar Si. In the case of $Co_3Se_4$ on planar Si, the current density started to increase negatively much earlier than that of bare planar Si. This indicates that the $Co_3Se_4$/Si system has higher onset potential, defined as the specific potential at $-1$ mA/cm$^2$. The onset potential ($V_{os}$) of $Co_3Se_4$/Si ($V_{os}$ = 0.216 V vs. RHE at $-1$ mA/cm$^2$) is

303 mV higher than that of planar Si ($V_{os}$ = −0.087 V vs. RHE at −1 mA/cm$^2$). The potential versus current plot under dark condition shows a 1 μA/cm$^2$ scale of current density, which indicates that the photoresponse of the $Co_3Se_4$/Si cell operated well with and without illumination. However, when $Co_3Se_4$ was deposited on the Si electrode, the saturation current density decreased by approximately 23% from the initial current density (from −34.1 to −26.4 mA/cm$^2$). This was attributed to the reduced light collection caused by deposition of the semi-transparent $Co_3Se_4$ film. To evaluate the photocathodic efficiency, a half solar-to-hydrogen conversion efficiency was introduced. It was defined as [(photocurrent density at the reversible potential (0 V vs. RHE), $J_{V=0}$ (mA/cm$^2$) × (open circuit potential, $V_{oc}$ (V) × Fill Factor/100 (mW/cm$^2$) × 100 (%)] [2,34]. The equation is also equal to a maximum value of [$J_{ph}$ (mA/cm$^2$) × ($V_{redox}$ − $V_b$) (V)/($P_{light}$ (mW/cm$^2$) × 100%)] [2,34]. Herein, $J_{ph}$ is the photocurrent density obtained under the applied bias $V_b$ and $V_{redox}$ is the redox potential for hydrogen production (0 V vs. RHE). $V_{bias}$ is an externally applied bias potential often necessary to achieve reasonable photocurrents and $P_{light}$ is the intensity of the incident light under AM 1.5 G conditions. In this work, the incident light intensity was maintained at 100 mW/cm$^2$. From calculation, the half solar-to-hydrogen conversion efficiency increased to 10-times higher than for bare planar Si. Consequently, it was determined that $Co_3Se_4$ is a highly active film for photoelectrochemical hydrogen production in terms of photocathodic efficiency.

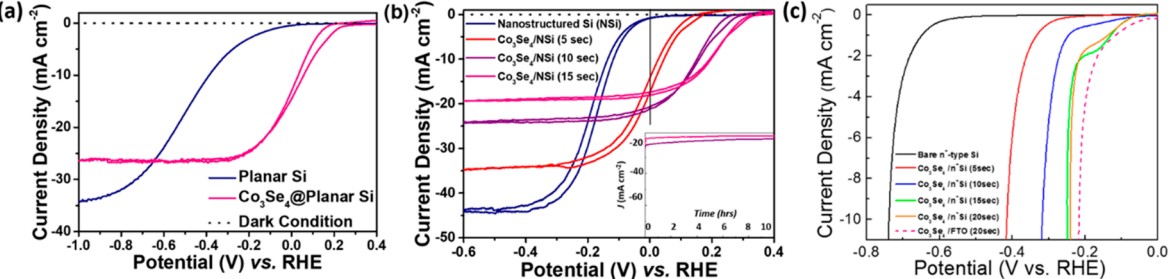

**Figure 2.** Cyclic voltammograms of electrodeposited $Co_3Se_4$ on Si photocathodes: (**a**) Current density-potential curves of bare planar Si photoelectrode and electrodeposited $Co_3Se_4$ on planar Si photoelectrode under light condition (solid lines). For comparison, $Co_3Se_4$ on Si under dark condition was also displayed (dashed line). (**b**) Photocurrent density-potential curves according to electrodeposition time of $Co_3Se_4$ under light illumination condition. Inset image indicates chronoamperometry results from selected photocathode samples for the stability test. $Co_3Se_4$/NSi (15 s) at the potential 0.2 V vs. RHE (pink color) and $Co_3Se_4$/NSi (15 s) at the potential 0.1 V vs. RHE (purple color). (**c**) Dark current density-potential curves of $Co_3Se_4$ film on highly P-doped n$^+$-type Si electrodes with increasing electrodeposition time of $Co_3Se_4$.

Treatment of the nanostructured Si surface using a metal-catalyzed electroless etching process further increased the light-to-hydrogen conversion efficiency. The FE-SEM images in Figure 3 portrays that the growth of nanostructured Si increases with respect to the electroless etching time. Optimized nanostructuring of the Si electrode surface (the sample with etching time of 20 min shown in Figure 3b) increased the limiting current density (the saturation current density at high negative potential) to 44.0 mA/cm$^2$ when deposited with $Co_3Se_4$ film. This is 1.29 times higher than that of bare planar Si (34.1 mA/cm$^2$), as shown in Figure 2a,b. The $V_{os}$ of the nanostructured Si (−0.007 V vs. RHE) also shifted by a positive 80 mV ($V_{os}$ of bare planar Si: −0.087 V vs. RHE). Consequently, the conversion efficiency of the nanostructured Si photocathode (0.04%) increased two fold compared to that of bare planar Si (0.02%). The enhanced efficiency of the nanostructured photoelectrode surface is mainly attributed to increased effective surface area and reduced reflectance of incident light. To increase the photocathodic activity even more, $Co_3Se_4$ was deposited electrochemically as a function of time, maintaining the applied potential of −0.8 V versus RHE. Figure 2b shows representative photoresponse results with various electrodeposition times. With increasing deposition time, the onset potential

shifted positively (anodically) while the limiting current density continuously decreased, induced by deposition of much thicker $Co_3Se_4$ film. Overall, the conversion efficiency increased and conversion became saturated at 15 s of electrodeposition time. Combined with the surface structure of $Co_3Se_4$ film, the very similar structures also resulted in the almost same light current density among samples of 15 s, 20 s and 25 s of electrodeposition time, which means that saturated porosity from the 3D structure showed the saturated light current densities according to the electrodeposition time. Consequently, the efficiency of the $Co_3Se_4$/nanostructured Si with 15 s of electrodeposition was 2.71%, a value 136 times higher than that of bare planar Si and 68 times higher than bare nanostructured Si without co-catalyst (Table 1). Moreover, Chronoamperometry test was implemented to measure the stability of selected $Co_3Se_4$/NSi (10 s) and $Co_3Se_4$/NSi (15 s) photocathode samples. The potential was applied approximately at its maximum current point (0.1 V vs. RHE for the 10 s sample and 0.2 V vs. RHE for the 15 s sample) during long-term operation. As a result, both samples showed a current density of approximately $-19$ mA cm$^{-2}$ and $-13$ mA cm$^{-2}$, respectively, which also maintained more than 80% of the initial current even after 10 h. From these results, it was shown that an optimal thickness of $Co_3Se_4$ co-catalyst film exists for efficient hydrogen production.

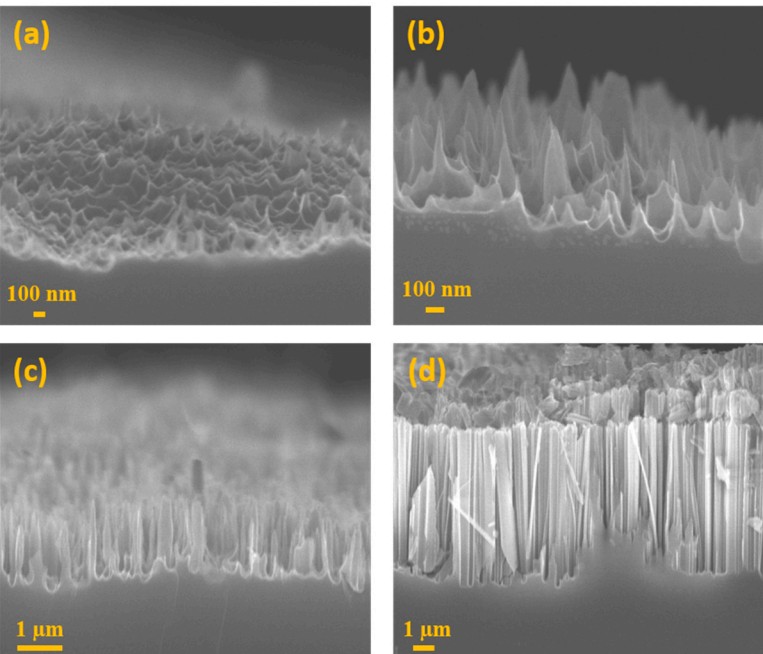

**Figure 3.** FESEM images Si photocathodes with variation of etching time (**a**) 10 min, (**b**) 20 min, (**c**) 40 min and (**d**) 150 min.

**Table 1.** Half solar-to-hydrogen conversion efficiency of various photocathodes.

| Type of Si Photocathode | Efficiency (%) |
| --- | --- |
| Planar Si | 0.02 |
| $Co_3Se_4$ on planar Si | 0.20 |
| Nanostructured Si | 0.04 |
| $Co_3Se_4$ on nanostructured Si (5 s) | 0.47 |
| $Co_3Se_4$ on nanostructured Si (10 s) | 1.72 |
| $Co_3Se_4$ on nanostructured Si (15 s) | 2.71 |

Possible native oxide formation at the interface between Si and $Co_3Se_4$ film might affect the photoelectrochemical performance. In terms of band structure theory, previous study and our measurement have shown that conduction band and valence band edges for bare Si electrode are approximately +0.5 V and $-0.6$ V versus RHE, respectively. In this system, $Co_3Se_4$ acts as the

co-catalyst, not acts as the photon-absorbing materials because we also measured photoresponse test with $Co_3Se_4$ film on FTO under light condition and confirmed that there was no any photocurrent density. The native oxide with a thickness of less than 5 Å can be formed on to the Si surface and can affect the photoelectrochemical property [35], however, the removal of the native oxide was also tried through HF treatment prior to electrodeposition process of $Co_3Se_4$ film. Consequently, the possibility of the formation of a very thin native oxide layer might be much minimized and a thickness of much less than 5 Å between Si and $Co_3Se_4$ is able to be insignificant to produce an affective energy barrier. Nevertheless, the formation of native oxide cannot be removed perfectly. The generated electron charge carriers in the depletion region of Si by incident light can relative easily move to the $Co_3Se_4$ co-catalyst to participate in the proton reduction to hydrogen production through the possible tunneling mechanism [36]. In addition, the remaining thin native oxide layer might be acted as the passivation layer during photoelectrochemical process combined with electrodeposited $Co_3Se_4$ film [37]. Similarly, band bending effect is the other possible effect contributed by SiOx layer formation. The photons absorbed by the silicon wafer generates minority carriers that can be diffused onto the $Co_3Se_4$ and band-bending effect can be accelerated by the native SiOx barrier layer. The formation of possible additional surface trap sites can be also suppressed by the native oxide formation at the interface. To study the exact effect of SiOx layer, we are under investigation to study these concepts in detail through in-depth studies such as Mott-Schottky plot by capacitance measurement or the change of carrier densities by the work function measure from UPS analysis in our next project.

To evaluate the electrocatalytic properties of the material, the HER electrocatalytic activity was examined using the customary three-electrode system. To understand better the electrocatalytic activity with respect to the substrate, the $Co_3Se_4$ material was deposited on GC and on FTO substrate. In Figure 4a, the HER polarization curves of $Co_3Se_4$ on FTO and GC electrodes are compared, along with those of bare FTO and bare GC. In all the polarization curves, the reductive current density started to increase negatively according to the cathodic potential applied and promoted the hydrogen evolution reaction (HER). The HER curves indicate that $Co_3Se_4(10)$ on the FTO electrode exhibited superior onset potential ($-0.189$ V vs. RHE at $-1$ $mA/cm^2$) than did $Co_3Se_4(10)$ deposited on GC ($-0.192$ V vs. RHE at $-1$ $mA/cm^2$) and better onset potential than did bare FTO ($-0.552$ V vs. RHE at $-1$ $mA/cm^2$) and GC ($-0.644$ V vs. RHE at $-1$ $mA/cm^2$) electrodes. Moreover, at a similar current density of 5 $mA/cm^2$, the $Co_3Se_4$ deposited on the FTO conductive electrode showed enhanced electrocatalytic activity resulting in a lower overpotential of 239 mV compared to $Co_3Se_4$ on the GC electrode (292 mV). This indicated that FTO is a more suitable substrate and improves catalytic activity. Therefore, in-depth analyses were carried out at various deposition times of the $Co_3Se_4$ electrocatalysts on the FTO substrate. Figure 4b shows that the HER activity of the $Co_3Se_4$ electrocatalyst varies with respect to the deposition time (5–25 s). The electrocatalytic activity of the time dependent $Co_3Se_4$ catalysts was observed to increase gradually with deposition time from 5 to 20 s, exhibiting a very low onset potential ($-0.205$ V to $-0.119$ V vs. RHE at $-1$ $mA/cm^2$). Subsequently, the sample deposited for 25 s abruptly declined due to an increase in onset potential ($-0.152$ V vs. RHE at $-1$ $mA/cm^2$) compared to the sample deposited for 20 s. Figure 4c displays the chronoamperometry (CA) curve of the $Co_3Se_4$ electrocatalyst carried out at a steady potential of $-0.2$ V (vs. RHE) for 7200 s. The evolution of $H_2$ gas bubbles from the surface of the electrode was noticed all over the CA analysis. Hence, the steady-state curve obtained from the CA scrutiny substantiates the highly stable nature of the prepared $Co_3Se_4$ electrocatalyst for HER activity in acidic $HClO_4$ medium. The inset of Figure 4c sorts out the essential overpotentials demanded by prepared $Co_3Se_4$ electrocatalysts deposited on the FTO for different times. The enhanced HER catalytic activity of the $Co_3Se_4$ (20 s) catalysts can be attributed to better quality loading of the catalyst on the FTO substrate. However, the decrease in the activity of the $Co_3Se_4$ (25 s) catalysts may be due to overloading of the catalysts on the FTO. This clumps the electroactive sites, resulting in a poor interface between electrode and electrolyte. Therefore, controlled loading of the catalysts over the substrates plays a vital role because it tends to generate more electroactive sites, which facilitates superior electrocatalytic activity. Overall,

the $Co_3Se_4$ catalysts deposited on the FTO substrate for 20 s delivered superior HER electrocatalytic activity by requiring only a very low overpotential of 215 mV to acquire an improved current density of 10 mA/cm$^2$. This overpotential (215 mV) required by the $Co_3Se_4$ catalysts electrodeposited on the FTO substrate for 20 s seems very efficient compared to the previously reported results on metal selenides.

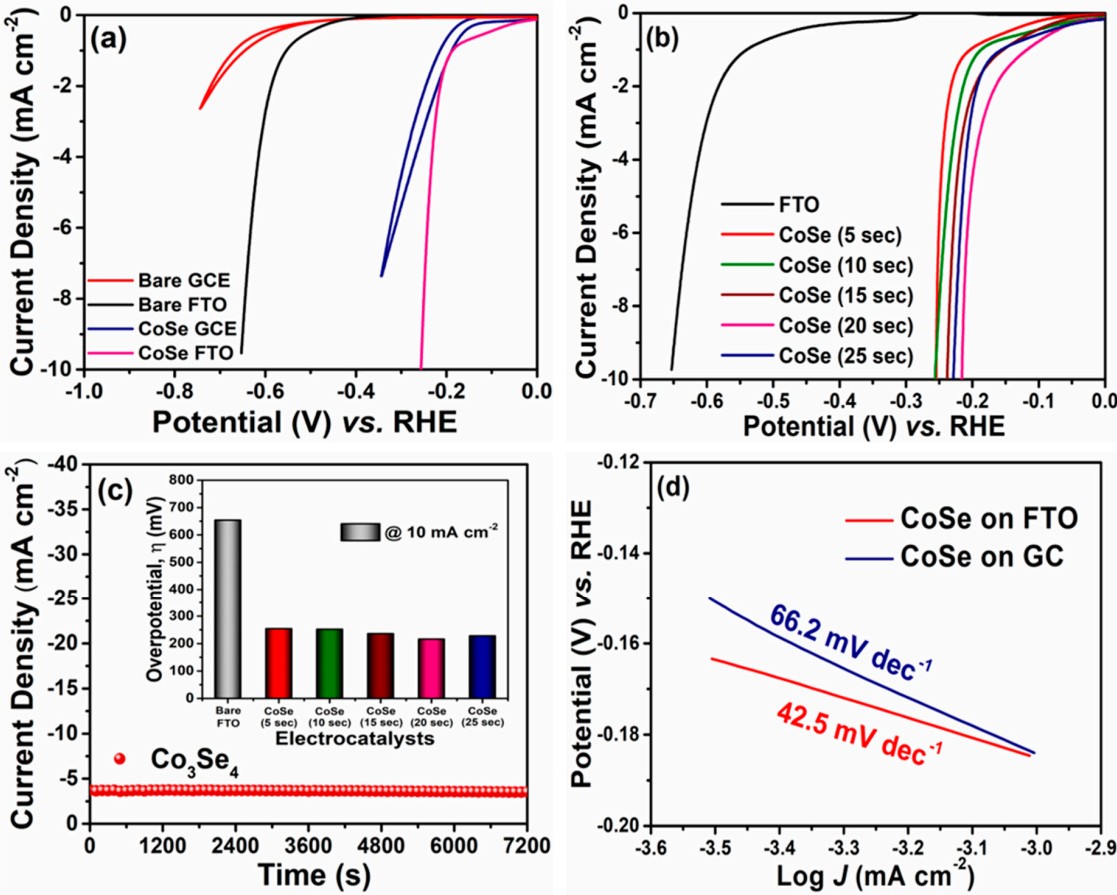

**Figure 4.** Electrochemical activity of cobalt selenide ($Co_3Se_4$) in 1 M $HClO_4$ with respect to deposition time on glassy carbon and FTO electrode from RDE system: (**a**) Polarization curves of bare GCE, bare FTO, $Co_3Se_4$(10) on GCE and $Co_3Se_4$(10) on FTO. (**b**) Polarization curves of bare FTO without any co-catalyst and FTO with cobalt selenide deposited for 5, 10, 15, 20 and 25 s. (**c**) the stability of electrodeposited $Co_3Se_4$ during 7200 s at a steady potential of $-0.2$ V (vs. RHE) (**d**) HER Tafel slope of glassy carbon and FTO electrodeposited with cobalt selenide.

To study the effect of $Co_3Se_4$ catalyst on to the different electrode substrate, the $Co_3Se_4$ film was deposited onto the Si substrate to compare electro-catalytic property under dark condition. In this study, highly P-doped n$^+$-type Si was selected for the electrochemical proton reduction to generate hydrogen (Figure 2c) and it is the general method to investigate electro-catalytic performance [38]. As a result, electrochemical behaviors were very similar to those of performance from $Co_3Se_4$ films on FTO substrate with increasing Co3Se4 deposition time (Figure 4). A slightly higher anodic shift in FTO samples might be attributed to the difference between the intrinsic resistivity or the conductivity of Si and FTO substrates. To sum up, in terms of electro-catalytic behavior, the same tendency of $Co_3Se_4$ co-catalyst film was identified from FTO and Si electrode substrate.

To gain additional and more detailed information about the inherent properties of the electrodeposited $Co_3Se_4$, the current density versus potential curves of the $Co_3Se_4$ samples on GC and on FTO were converted to Tafel plots to study the rate limiting step of the HER reaction. First, the Tafel slopes shown in Figure 4d were calculated to be 66.2 and 42.5 mV/dec for $Co_3Se_4$ on GC

and on FTO, respectively. The $Co_3Se_4(20)$ electrocatalyst deposited on FTO yielded a consistently low Tafel value, substantiating the upgraded electrocatalytic activity of the material. This provided further evidence that hydrogen can be generated by a Heyrovsky reaction mechanism, by which one proton adsorbs to the electrode and one hydrated proton comes from the electrolyte (and they combine) to make hydrogen gas. The exchange current density of $Co_3Se_4$ on GC was $1.62 \times 10^{-6}$ A/cm$^2$ and that of $Co_3Se_4$ on FTO was $4.47 \times 10^{-8}$ A/cm$^2$, which values were derived from the results on variation in the substrate condition. In conclusion, electrodeposited $Co_3Se_4(20)$ showed an effective hydrogen evolution mechanism with any of the conductive standard electrodes.

To measure the electrochemically active surface area (ECSA), the capacitance of an electrical double layer ($C_{dl}$) at a solid/electrolyte interface was evaluated from the CV curves of $Co_3Se_4$ on GC and FTO at various scan rates, as shown in Figure 5. The $C_{dl}$ is estimated from the slope of the plot of $J_c$, which is the current at the constant potential with increasing scan rate from 20 to 100 mV/s. The capacitance is 0.19 mF/cm$^2$ and 0.66 mF/cm$^2$ for $Co_3Se_4$ on GC and FTO, respectively. These values are much higher than that of a typical compact flat electrode reported to date (approximately 10–20 μF/cm$^2$) [39]. The slope is also proportional to the exchange current density, which is directly related to the electrocatalytically active surface area [25,40]. The ECSA is calculated from the $C_{dl}$ divided by the capacitance of a smooth planar surface of the catalyst ($C_s$). Using the general $C_s$ value of 13–17 μF/cm$^2$ in pH 0 solution (1M HClO$_4$) [41], the average ECSA of $Co_3Se_4$ on GC and FTO are $(2.67 \text{ and } 4.45) \times 10^{-5}$ cm$^2$, which corresponds to roughness factors of $(13.67 \text{ and } 22.25) \times 10^{-4}$ cm$^2$, respectively. Therefore, $Co_3Se_4$ on GC has a large electrochemical catalytic surface area, which may also provide effective active sites for the HER.

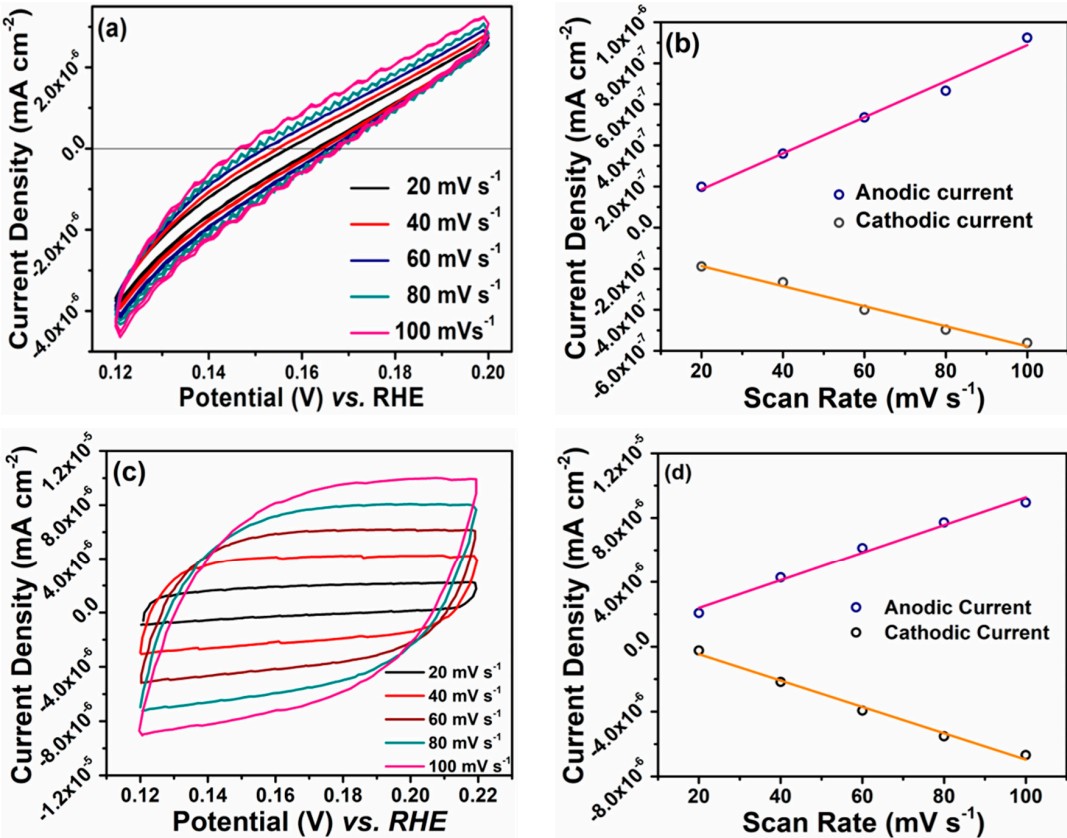

**Figure 5.** The current vs. potential result at diverse scan rates and The capacitance current ($J_C$). (**a**) The CV curve of $Co_3Se_4$ on GC with different scan rate. (**b**) $J_c$ of $Co_3Se_4$ on GC at $J_{net} = 0$ mA cm$^{-2}$ depending on scan rates. (**c**) The CV curve of Co3Se4 on FTO with different scan rate. (**d**) $J_c$ of $Co_3Se_4$ on FTO at $J_{net} = 0$ mA cm$^{-2}$ depending on scan rates.

## 4. Conclusions

In conclusion, we employed electrodeposited-cobalt selenide ($Co_3Se_4$) co-catalyst on a silicon photoelectrode system for the water splitting reaction. The SEM image confirms the uniform distribution of the $Co_3Se_4$ film over the electrode material. The controlled loading of $Co_3Se_4$ over the substrates was effectively optimized by varying the deposition time. The prepared $Co_3Se_4$ electrodes were successfully investigated and found to be efficient electrocatalysts for hydrogen evolution reactions. For the hydrogen evolution reaction, the $Co_3Se_4$ catalyst electrodeposited on FTO substrate for 20 s requires very low overpotential of 215 mV to provide an improved current density of 10 mA/cm$^2$, compared to the other prepared electrocatalysts. Moreover, this electrocatalyst yielded a very low Tafel slope value of 42.5 mV/dec. From these enhanced HER catalytic activity results, we believe that $Co_3Se_4$ prepared by this easy, one-pot electrodeposition method, seems to be the most promising candidate for commercialization as a low cost efficient electrocatalyst for water splitting.

**Author Contributions:** Conceptualization, U.S.; Methodology, U.S.; Validation, U.S.; Formal Analysis, Y.S. and S.S.; Investigation, S.S. and U.S.; Resources, U.S.; Data Curation, B.M. and S.S.; Writing-Original Draft Preparation, Y.S., J.J and S.S.; Writing-Review & Editing, U.S., J.J., S.S. and Y.S.; Visualization, B.M. and Y.S.; Supervision, U.S.; Project Administration, U.S. † These authors contributed equally to this work.

**Funding:** This research was supported by the National Research Foundation of Korea (NRF) and funded by the Ministry of Science, ICT & Future, Republic of Korea (2018R1C1B6001267 and 2018R1A5A1025224).

**Conflicts of Interest:** The authors declare no conflict of interest.

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
