# Peer review of "Efficient Photoelectrochemical Water Splitting Reaction using Electrodeposited Co3Se4 Catalyst"

_applsci, doi:10.3390/app9010016_

Round 1

Reviewer 1 Report

In the manuscript the authors suggest Co3Se4 as a possible alternative protection and HER layer for PEC water splitting. While the manuscript is generally interesting a few additional experiment results should be included to really indicate the suitability of the prepared layers. Also the use of Co3Se4 as an OER material (at strength conditions, pH 8) is rather distractive and interfere with the general massage of the manuscript. So it is encourage to focus the manuscript as suggested by the title to PEC applications. Points that should be addressed:

-          An important point is stability of the prepared devices. This should be convincingly addressed in the manuscript, e.g. by extended measurements or determination of FE

-          Characterization of the nanostructured Si should be provided, e.g. SEM

-          In the experimental section it should be mentioned how efficiencies were calculated and a description of the standard protocol for sample cleaning should be included.

-     The experimental conditions should be stated in the figure captions, e.g. electrolytes used

-          A few relevant papers are missing in the introduction and in the general discussion of the  results. The authors should include the following articles: DOI: 10.1021/acs.jpclett.5b00495 (comparison planar vs microwire PEC with CoP HER catalysts), DOI: 10.1039/C5EE02188J (Limits of Pt HER catalyst in PEC), 10.1039/C6EE02215D and 10.1002/aenm.201400739 (alternative non-noble metal catalytic coatings)  

-          The experimental conditions seem to be not appropriate to determine the ESCA. In fact it is not really clear what the purpose of the measurements is as the comparison with FTO samples is missing. For Tafel slope and exchange current density calculations the precise surface area should have been used.

Author Response

We appreciate your kind consideration and the reviewers for providing their valuable comments/suggestions to improve the quality of our recent submission for the consideration of publication by Applied Sciences (Manuscript ID: applsci-386504). According to the reviewers’ comments, we have thoroughly revised the paper. We hope this revision can meet the reviewers’ and your requirements. The followings are our point-by-point responses: 

Reviewer 2 Report

I think this paper is well written and interesting.

I have some suggestions:

The figure numbering is wrong

 line 195: figure 2a should be figure 3; line 212 figure 2 should be figure 3.

The SEM images of samples deposited on fro and GC should be added to have a better idea of their morphology and to understand better the difference between the sample deposited for 20s and that deposited for 25s.

The authors did not say anything about the junction formed between the SiO2/Si and Co2SeO4. What is the relative positions of CB and VB edges for Si and Co3Se4. What is the effect of the SiOx barrier?

I also don't think that the behavior of Co3Se4 on FTO is identical to that on Si. The authors should discuss this aspect in greater detail.

Author Response

(The authors gave the same response as above.)

Round 2

Reviewer 1 Report

While the manuscript has certainly improved since the initial submission the stability of the device, i.e. the photocathode has not been convincingly addressed. The authors must provide stability data of the photocathode operated at its max. power point (0.2V vs RHE for Co3Se4/NSi(15 sec), 0.1V vs RHE for Co3Se4/NSi(10 sec)). Currently the stability measurement is meaningless as the substrate (not even specified) might not corrode during testing. Si however will corrode/passivate upon contact with the electrolyte, i.e. in case porous Co3Se4 films are deposited. Along with a chronoamperometric (CA) measurement (2hours are not sufficient as state-of-the-art devices operate for > 100h) a CV after CA measurement should be shown.

Additionally, the authors removed the OER part but it is still mentioned in the introduction and conclusion. So revision is certainly required.

Author Response

Please see the attached response letter.

Reviewer 2 Report

The authors have improved the manuscript andd answered all the question and issues raised in my previous review. This paper can be published in the present form.

Author Response

I really appreciate it. Thank you so much for your response and comment.

Round 3

Reviewer 1 Report

I have no further comments.